

# Comparing different farming habitats for mid-water rope nurseries to advance coral restoration efforts in the Maldives

Inga Dehnert[1,2], Luca Saponari[3], Paolo Galli[1,2] and Simone Montano[1,2]

[1] Department of Earth and Environmental Sciences (DISAT), University of Milan-Bicocca, Milan, Italy
[2] MaRHE Center (Marine Research and High Education Center), Magoodhoo Island, Faafu Atoll, Republic of Maldives
[3] The Centre for Environment & Education, Nature Seychelles, Mahe, Republic of Seychelles

## ABSTRACT

The need for comprehensive and effective coral restoration projects, as part of a broader conservation management strategy, is accelerating in the face of coral reef ecosystem decline. This study aims to expand the currently limited knowledge base for restoration techniques in the Maldives by testing the performance of mid-water rope nurseries in a lagoon and a reef habitat. We examined whether different coral farming habitats impacted fragment survival, health and growth of two coral genera and how the occurrence of mutualistic fauna, predation and disease influenced coral rearing success. Two nurseries were stocked with a total of 448 *Pocillopora verrucosa* and 96 *Acropora* spp. fragments, divided into different groups (four *Pocillopora* groups: lagoon nursery at 5 m; reef nursery at 5, 10 and 15 m; two *Acropora* groups: lagoon nursery at 5 m and reef nursery at 5 m). Eight fragment replicates from the same donor colony (*Pocillopora* genets: $N = 14$, *Acropora* genets $N = 6$) were used in each group and monitored for one year. Our results show that fragment survival was high in both farming habitats (>90%), with *P. verrucosa* surviving significantly better in the lagoon and *Acropora* spp. surviving and growing significantly faster in the reef nursery. *P. verrucosa* growth rates were similar between reef and lagoon habitat. Different rearing depths in the reef nursery had no impact on the survival of *P. verrucosa* but coral growth decreased considerably with depth, reducing fragments' ecological volume augmentation and growth rates by almost half from 5 to 15 m depth. Further, higher fish predation rates on fragments were recorded on the reef, which did not impact overall nursery performance. Mutualistic fauna, which correlated positively with fragment survival, was more frequently observed in the lagoon nursery. The occurrence of disease was noted in both habitats, even though implications for fragment health were more severe in the lagoon. Overall, our study demonstrates that lagoon and reef nurseries are suitable for rearing large numbers of coral fragments for transplantation. Nevertheless, we recommend considering the specific environmental conditions of the farming habitat, in particular water quality and year-round accessibility, in each case and to adjust the coral farming strategy accordingly. We hope that this novel research encourages the increased application of mid-water rope nurseries for 'coral gardening' to advance coral reef recovery and climate resilience in the Maldives.

Corresponding author
Inga Dehnert,
inga.dehnert@unimib.it,
ingadehnert@gmail.com

## INTRODUCTION

Coral reef restoration has become an increasingly applied tool and internationally adapted approach to counteract the worldwide degradation of coral reefs (*United Nations Environment Assembly, 2019*; *Boström-Einarsson et al., 2020*). While sometimes criticized for not tackling the underlaying problem and therefore using limited conservation resources inefficiently (*Bellwood et al., 2019*; *Morrison et al., 2020*), supporters argue that, in concert with other environmental measures, rigorously managed local restoration projects can improve social, economic and ecological resilience, and therefore increase the odds for reef survival and recovery (*Hein et al., 2019*; *Hein et al., 2021*; *Duarte et al., 2020*). Such projects may also prove valuable in the face of global threats that are often beyond the level of local or even national control.

The low-lying archipelago of the Maldives, a country that owes its existence to the 26 natural coral atolls, is on the forefront of experiencing the adverse effects of climate change. Over the next decades, the nation's mere existence will depend on its ability to protect its population, infrastructure, economy and coral reef ecosystem from the risks posed by warming oceans, sea level rise and severe weather events (*Sovacool, 2012*; *Storlazzi et al., 2018*). Maldivian coral reefs are essential for the country's economy, that heavily relies on tourism and fisheries (*Statistical Yearbook of Maldives, 2020*). Nevertheless, Maldivian reefs have already seen considerable degradation following several mass bleaching events (*Tkachenko, 2015*; *Perry & Morgan, 2017*) along with other threats such as pollution, corallivores and disease outbreaks (*Jaleel, 2013*; *Montano et al., 2016*; *Saponari et al., 2018*; *Montalbetti et al., 2019*). Monitoring data from the most recent mass-bleaching in 2016 reported that 73% of shallow water corals were bleached across the Maldives (*Ibrahim et al., 2017*). Subsequent changes in Maldivian coral community structure included the disproportionately high mortality of reef-building *Acropora* species as well as an observed shift from mature populations towards small and medium sized colonies (*Pisapia, Burn & Pratchett, 2019*). Preserving and restoring the resilience of Maldivian coral reefs, through environmental protection and active restoration should therefore be of immediate priority to brace the archipelago against climate change. After all, healthy and structurally complex reefs can, for example, provide protection against coastal erosion (*Harris et al., 2018*) and may even help islands to grow upwards in response to sea level rise (*Masselink, Beetham & Kench, 2020*).

In the past, restoration projects in other locations have demonstrated the ability to mitigate the continued degradation of coral reefs. For example, large scale, long-term reef restoration was successfully conducted in Indonesia, following physical reef degradation from blast fishing and other human activities. Coral cover increased significantly following rehabilitation treatment to stabilize substrate in Komodo National Park (*Fox et al., 2019*) and the deployment of artificial structures with attached coral fragments increased not only live coral cover by more than 50%, but also demonstrated minimal subsequent bleaching impacts despite warm waters and continued disturbances (*Williams et al., 2019*).

Surprisingly, coral reef restoration activities are not widely applied in the Maldives. Peer-reviewed studies of direct transplantation and concrete blocks as artificial reef structures

date back to the 1990s (*Clark & Edwards, 1994*; *Clark & Edwards, 1995*). Currently, the dominant form of restoration appears to be the application of metal frames, also known as spiders, as artificial reefs, a practice that can be easily applied in a resort setting and also serves as an educational tool (*Edwards, 2010*; *Hein et al., 2019*). However, larger active restoration projects applying the 'gardening concept' of a farming and an outplanting phase (*Rinkevich, 1995*; *Rinkevich, 2000*; *Epstein, Bak & Rinkevich, 2001*) are relatively uncommon and undocumented, especially in community or resort-based projects.

Mid-water floating nurseries, and rope nurseries in particular, allow small, fragmented corals to grow fast under optimal conditions due to increased light and water flux, reduced sedimentation and overgrowth as well as protection from demersal predators (*Shafir, Van Rijn & Rinkevich, 2006*; *Levy et al., 2010*). They have proven an effective tool in gardening projects around the world in order to increase fragment survival and growth while continuously building a bigger re-sourcing and farming stock (*Shafir & Rinkevich, 2010*; *Frias-Torres, Montoya-Maya & Shah, 2018*; *Bayraktarov et al., 2020*).

When deciding on the in-situ nursery location, it is recommended to consider water quality, depth, shelter and accessibility while also aiming for similar environmental conditions of the targeted transplantation site (*Frias-Torres, Montoya-Maya & Shah, 2018*). Therefore, nurseries are often placed in shallow lagoons, where the growing fragments are protected from the forces of currents and weather as well as corallivorous reef predators (*Levy et al., 2010*). The nurseries soon turn into floating ecosystems by attracting fish assemblages which can reduce cleaning requirements and costs as they consume biofouling (*Shafir, Van Rijn & Rinkevich, 2006*; *Shafir & Rinkevich, 2010*). However, reef environments typically already host diverse fish communities that could provide cleaning services or even pose a predation risk (*Frias-Torres & Van de Geer, 2015*; *Seraphim et al., 2020*). On the reef, environmental conditions are also more likely to resemble the future transplantation site, while nursey structures are more exposed to natural forces and likely more difficult to construct.

Selecting a suitable rearing environment is therefore a crucial factor for the success of any coral gardening project and requires careful, knowledge-based assessment. In the Maldives, coral reefs and their lagoon habitats cover approximatively 20% of the county's Territorial Sea (*Naseer & Hatcher, 2004*). Yet potential nursery sites may vary considerably in their characteristics and decision driving evidence remains to be verified for this part of the Indian Ocean.

This study provides an in-depth comparison of the performance of mid-water rope nurseries in a lagoon and reef habitat in the Maldives over a one-year monitoring period for the first time. We assessed the survival, health and growth of the same genotypes of *Pocillopora verrucosa* and *Acropora* spp. fragments to better understand the positive and negative implications of these farming environments and their specific challenges. With our findings we hope contribute to the informed decision making in active restoration projects and encourage the wider application of this technique in the Maldives, particularly in tourist resort settings.

## MATERIALS & METHODS

### Study design

This study assessed coral nursery farming performance in two habitats, an inner atoll reef and a sheltered lagoon environment, on Athuruga Resort Island (3°53′14″N72°48′59″E) in Alif Dhaal atoll, in the Republic of Maldives (Fig. 1A). Two mid-water rope nurseries, one in each location, were simultaneously stocked in February 2020 and fragment development was monitored for one year. The lagoon nursery (LN) was situated away from daily resort activities, about 500 m from the main island, anchored at 10 m depth and comprised horizontally suspended 10 m long coral ropes attached to PVC pipes at 5 m depth (Fig. 1B). Athuruga's large lagoon, measuring approximately 1,200 m from west to east and 650 m from north to south, is surrounded by a reef rim and only connects via a narrow artificial channel to the inner Atoll Sea. No currents are experienced here and visibility is typically low. The lagoon floor is characterized by a sandy bottom with an abundant echinoderm fauna, in particular various sea cucumber species and large seastars such as the corallivorous *Culcita* sp. The isolated reef patches that, following the 2016 mass bleaching, mainly comprise of dead corals and some living *Porites* colonies concentrate the limited fish life. The reef nursery (RN) was placed parallel to the island's southern house reef, that exhibits a steep slope in this area. Here, the once abundant and diverse live coral cover has also been severely reduced to less than 5%, following the latest bleaching and an outbreak of the corallivorous seastar *Acanthaster planci* (*Saponari et al., 2018*; *Saponari et al., 2021*), with some larger massive coral colonies and a comparably abundant reef fish community remaining. The RN was anchored at 20 m depth, about 5–10 m away from the reef slope and a more streamlined design (no PVC pipes) was chosen to account for the increased exposure to slight to moderate currents on the reef. Horizontally suspended, 14 m long coral ropes were directly attached to the vertical anchoring ropes at three different depths (5, 10 and 15 m) (Fig. 1C).

For the purpose of this study, the two nurseries were stocked with a total of 544 experimental fragments from two coral genera, namely *Pocillopora* and *Acropora*.

*Pocillopora verrucosa* fragments derived from 14 donor colonies (12–18 cm diameter) that were previously reared in the two mid-water rope nurseries on Athuruga (hereafter referred to as lagoon or reef 'donor farming habitat' of experimental fragments). These donors were originally collected in 2018 from two natal sites with similar conditions to Athuruga's farming habitats. *Pocillopora* donors growing in the reef nursery originated from artificial substrate on Athuruga reef (*i.e.,* mooring lines) and were reared between 7 and 18 m depth. Donors growing in the lagoon nursery at 5 m were originally collected from the shallow back reef of Thudufushi Island (3°47′05″N72°43′49″E), since Athuruga lagoon did not offer sufficient live corals for nursery stocking. All donor colonies were assumed to be of different genotype as they were initially collected as corals of opportunity spaced more than 10 m apart (*Edwards & Gomez, 2007*; *Foster, Baums & Mumby, 2007*). In order to prevent any bias in nursery comparison resulting from possible habituation to the farming habitat or translocation to a different habitat, seven 'reef donor' and seven 'lagoon donor' colonies were used for the experiment (see Fig. S1 for experimental design graphic).
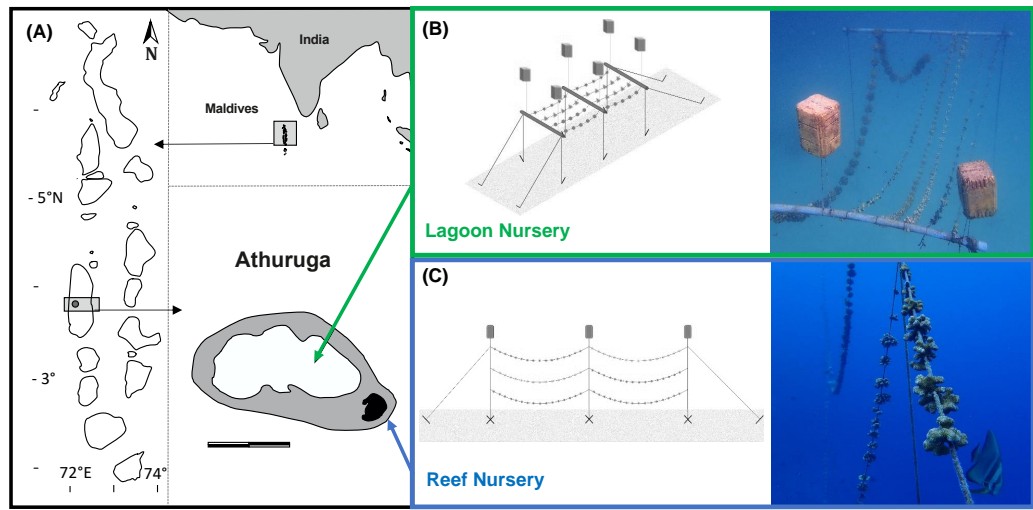

**Figure 1  Study location and mid-water rope nursery design.** (A) Map showing the Republic of Maldives, where Athuruga Resort Island (3°53′14″N 72°48′59″E) is located in the center of Alif Dhaal atoll (scale bar: 1 km; island in black, reef in grey, water in white). (B) Lagoon mid-water rope nursery (LN) adjusted from *Levy et al. (2010)* measuring 3 m in width and 10 m in length at coral rearing level at 5 m water depth. The main structure consists of 3 PVC pipes, connected with 10 mm rope to the anchoring iron bars and air-filled buoyancy containers pulling the structure upwards. (C) Reef mid-water rope nursery (RN) with adjusted streamline design, build parallel to the reef and anchored at 20 m depth. Coral ropes are attached at 3 different depth levels (5, 10 and 15 m).

To compare coral farming performance between the lagoon and the reef habitat and for different depths in the RN, a total of 448 *P. verrucosa* fragments were stocked, divided into four groups (Poc_LN_5m; Poc_RN_5m; Poc_RN_10m; Poc_RN_15m) according to nursery habitat and rearing depth. Each of the 14 donor colonies were fragmented in 32 similar sized fragments, ranging from 3–10 cm in diameter depending on the selected fragmentation size for each donor colony. Then, a subset of eight fragments was used for each study group, resulting in a total of 112 fragments per group with the same distribution of fragment genotypes and sizes. Fragmentation of *P. verrucosa* donor colonies from the RN and LN and restocking occurred on the nursery site and underwater using SCUBA equipment. To limit handling stress and damage, the stocked ropes were immediately reattached to the nurseries. Fragments that required translocation to a different rearing habitat were continuously submerged in separate containers and transported by divers the same day. Excess fragments were reared on separate ropes in the nurseries and excluded from the study.

*Acropora* fragments were directly collected as corals of opportunity from a nearby reef (3°48′51″N72°45′10″E) from less than 5 m unshaded depth. Six suitable colonies were selected based on their fragmentable size (15–20 cm in diameter), similar arborescent branching morphology and distance between them (>30 m) to increase genetic diversity. As available *Acropora* spp. fragments possibly comprise more than one species, all comparisons are made at the genus level. The donor colonies were kept in shaded and spacious containers filled with fresh seawater and transported to Athuruga within one hour, followed by the
same fragmentation and stocking procedure as for *P. verrucosa*. In the nurseries, *Acropora* spp. fragments represented two study groups of 48 fragments each, growing at 5 m depth in the LN and the RN (Acr_LN_5m; Acr_RN_5m). Again, subsets of 8 similar sized fragments (3–11 cm diameter) per donor colony were used, likely representing six different genotypes.

A monthly monitoring and maintenance protocol was established for a one-year farming period. The protocol was interrupted due to Covid-19 from months three to eight, resulting in a total of seven surveys (T1, T2 and T3–T7 post interruption) and three growth measurements at stocking (T0), post interruption (T3) and after one year (T7) for all fragments. Water temperature was recorded at 5 m depth during each dive using a Suunto dive computer.

## Data analysis

The status of nursery-grown experimental fragments was analyzed applying the following parameters suggested by *Frias-Torres, Montoya-Maya & Shah (2018)*: 'Survival' was determined as a binary condition ('alive' and 'dead') for each treatment group, habitat and genus and was compared using the chi-square test of independence. Fragment 'Condition' (see Fig. 2) was recorded as a categorical variable for each survey, distinguishing between fragments with 100% living tissue (H3), more than 50% living coral tissue (H2), and less than 50% living tissue on the fragment (H1) and is shown in percentage for each category and fragment group. It was further noted, whether fragments showed any signs of bleaching, disease or algae overgrowth. Predation incidents were recorded when fresh bitemarks or predation scars were evident on the fragments. The presence of any sessile corallivores, mutualists or any other visible fauna associated with the coral was also recorded. The percentage of fragments with diseased tissue was calculated for the last survey (T7), while associated fauna and predation rates were calculated as percentage of affected corals per study group for each survey and averaged across the study period. For predation and disease calculations dead fragments were excluded. Differences between habitats and depths as well as associations between mutualistic fauna and fragment survival were analyzed using the chi-squared test, with a post hoc residual analysis for different depth groups with a Bonferroni adjusted alpha level of 0.008 for the predation analysis.

Fragment initial size at stocking and 'Growth' was calculated for all fragments as 'Ecological volume' (EV) by taking three measurements to the nearest mm using a Vernier caliper, where:

$$EV = \pi r2h, \quad \text{where } r = (w + l)/4$$

with 'h' representing the longest linear colony diameter of the three perpendicular measurements ($h$ = height, $w$ = width, $l$ = length; see *Shafir, Van Rijn & Rinkevich, 2006*). The difference in EV at the start (T0) and the end (T7) of the study was compared using the Wilcoxon signed rank test and used to compute 'Size augmentation' and 'Daily growth rates' for all living fragments in each group. Growth rate data was natural log transformed to meet the homogeneity of variance assumption and analyzed using an ANOVA with Turkey's post hoc test.

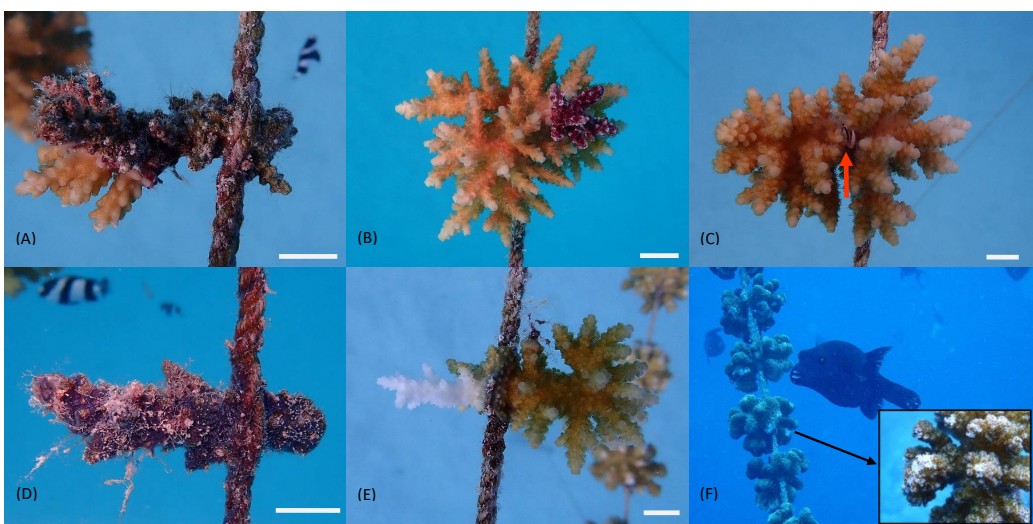

**Figure 2   Categories for coral fragment assessment.** (A) H1: less than 50% tissue alive. (B) H2: more than 50% tissue alive. (C) H3: 100% tissue alive; arrow indicates a guard crab *Trapezia* sp. (D) 100% mortality. (E) 'White syndrome' diseased fragment. (F) Fish predation and fresh predation marks. 1 cm white scale bars (photos by I. Dehnert).

In addition, the relationship between fragments' initial size (EV at T0) and the subsequent growth rate for *P. verrucosa* fragments was investigated using a Pearson correlation to obtain a better understanding of optimal stocking size for this species.

The experimental design further allowed to test for any differences between *P. verrucosa* fragments originating from 'reef reared' and 'lagoon reared' donors (*i.e.,* whether fragments from lagoon or reef reared donor colonies grew significantly different in the RN and the LN farming habitat). Therefore, mean differences in growth rates between fragments originating from reef and lagoon farming habitats were compared within each study group using the Mann–Whitney test.

All statistical analysis was performed using SPSS ver. 27 (IBM, New York) and all data is represented as arithmetic means ± standard error. Non-parametric test statistics were used when the normality assumption was violated.

## RESULTS

### Survival

Overall, the survival of the experimental stock ($N = 544$) was high (91%) after one year (T7) with differences between *Pocillopora verrucosa* (94%; $N = 448$) and *Acropora* spp. (89%; $N = 98$) fragment survival being marginally non-significant ($\chi^2(1, N = 544) = 3.59, p = 0.058$). For *P. verrucosa* the survival rate was above 90% for all four groups with the highest survival recorded in the LN (99%), which was significantly different from the RN survival ($\chi^2(1, N = 448) = 6.95, p = 0.008$). Here, the average survival rate for all depths was 92% and rearing depth had no significant effect on survival ($\chi^2(2, N = 336) = 1.334, p = 0.513$; see Table 1). In contrast, the survival of *Acropora* spp. fragments, all

Dehnert et al. (2022), *PeerJ*, DOI 10.7717/peerj.12874

**Table 1  Coral nursery performance of *Pocillopora verrucosa*.** The table shows fragment survival, disease incidents, predation and associated fauna rate (average rate of affected fragments per survey), ecological volume (EV) size augmentation and daily growth rates after a one-year farming period (T7 = 371 days) for the different study groups reared in mid-water rope nurseries in a lagoon (LN) and reef (RN) habitat at different depths.

| Group | Nursery habitat | Depth (m) | No. of fragments | Stocking period (days at T7) | Survival (% at T7) | Disease (% at T7) | Predation rate (mean% ± SE) | Fauna occurrence (mean% ± SE) | EV size augmentation (cm³ at T7 ± SE) | Daily growth rate (at T7 ± SE) |
|---|---|---|---|---|---|---|---|---|---|---|
| Poc_LN_5m | Lagoon | 5 | 112 | 371 | 99.11[**] | 0 | 37.19 ± 18.14 | 25.64 ± 6.83[***] | 863.60 ± 29.99 | 0.08 ± .005 |
| Poc_RN_5m | Reef | 5 | 112 | 371 | 90.18 | 2.97 | 58.64 ± 15.56[***] | 11.48 ± 3.10 | 739.74 ± 5.17 | 0.07 ± .004 |
| Poc_RN_10m | Reef | 10 | 112 | 371 | 91.07 | 5.88 | 48.66 ± 13.78 | 4.46 ± 1.28 | 539.14 ± 20.55 | 0.05 ± .003[***] |
| Poc_RN_15m | Reef | 15 | 112 | 371 | 94.64 | 1.89 | 39.75 ± 15.45[***] | 5.23 ± 1.51 | 386.31 ± 13.56 | 0.04 ± .002[**] |
| Poc_RN_total | Reef | all | 336 | 371 | 91.96 | 3.56 | 47.32 ± 13.50[***] | 7.06 ± 1.89 | 552.28 ± 14.24 | 0.05 ± .001 |
| Poc_all | all | all | 448 | 371 | 93.75 | 2.62 | 52.63 ± 15.45 | 11.70 ± 3.10 | 634.56 ± 14.73 | 0.06 ± .002 |

Significance levels: *** <0.001, ** <0.01 and *<0.05

growing at 5 m depth, was significantly higher in the RN (96%) than in the LN (81%; see Table 2) ($\chi^2(1, N = 96) = 5.031, p = 0.025$).

## Condition

Similarly, the majority of *P. verrucosa* fragments (RN: 88%; LN: 96%) were fully alive (H3) after one year (T7) with only a few partially alive corals (H2 and H1) found in each RN group ($N = 4$ at 5 m; $N = 6$ at 10 m and $N = 2$ at 15 m; see Fig. 3). In the LN only 3 fragments had suffered partial mortality (H2). No signs of disease were observed in *P. verrucosa* stock in the LN, while 3.6% of RN fragments were diseased with a rapid tissue loss syndrome (see *Moriarty et al., 2020*) at the last survey ($N = 3$ at 5 m, $N = 6$ at 10 m and $N = 2$ at 15 m; Table 1).

For *Acropora* spp., fragment health was more variable. In the RN 63% of the fragments were fully alive, while 33% had suffered partial mortality (H2 = 23%; H1 = 10%) due to algae overgrowth. In the LN, the spread of 'White Syndrome' disease (see *Montano et al., 2012*) had considerably impacted fragment condition (H2: 46%; H1: 35%; see Fig. 3) with no fully alive fragments remaining after one year and 18% of the living stock showing diseased tissue at T7, which was also the main cause of death in this group (Table 2).

On *P. verrucosa* fragments the average predation rate was significantly lower in the LN ($37 \pm 18\%$) than in the RN ($47 \pm 14\%$) ($\chi^2(1, N = 3481) = 13.504, p < 0.001$), where predation decreased significantly from 5 m (423 predation incidents in total) to 15 m depth (245 predation incidents; see Table 1) ($\chi^2(2, N = 2592) = 90.483, p < 0.001$). In the RN, predation events were also more consistent throughout the study period (in 6 out of 7 surveys), while in the LN predation on fragments was only recorded in three surveys. Predation on *Acropora* spp. was only recorded once on two fragments in the RN. Corals only showed fish predation marks in both habitats, which never made up more than 5% of the fragment's surface and visibly healed between surveys.

Of the fragment inhabiting fauna, guard crabs of the genus *Trapezia* were most frequently observed (90%; $N = 475$), while other small crabs, shrimps and fish made up the remaining 10%. Coral associated fauna was significantly higher in the LN ($\chi^2(1, N = 3584) = 193.24, p < 0.001$). Specifically, associated fauna was on average most frequently observed on *P. verrucosa* fragments in the LN ($26 \pm 7\%$), while only found in $7 \pm 2\%$ of RN fragments. Similarly, $23 \pm 5\%$ of *Acropora* spp. fragments in the LN were associated with fauna while in the RN it was only $4 \pm 2\%$. A significant positive relationship between *P. verrucosa* survival and *Trapezia* crabs occurrence was found ($\chi^2(1, N = 3584) = 9.674, p = 0.002$).

Temperature or stress induced bleaching was not an issue during the rearing period and water temperatures never exceeded 30 °C at 5 m depth in either habitat. Temporary bleaching of the upper fragment tissue was only observed in 3 fragments (1 Poc at LN; 1 Poc and 1 Acr in RN) during the study. Brown algae (*Sargrassum* sp.) overgrowth was most noticeable on *Acropora* fragments in the RN, where 10 fragments had suffered partial tissue damage at T3 due to the interrupted maintenance schedule. In contrast, blue–green algae, identified in the field as mainly *Schizothrix calcicola* were prevalent on the LN structure, but did not overgrow living fragments.

Dehnert et al. (2022), *PeerJ*, DOI 10.7717/peerj.12874

**Table 2  Coral nursery performance of *Acropora* spp.** The table shows fragment survival, disease incidents, predation and associated fauna rate (average rate of affected fragments per survey), ecological volume (EV) size augmentation and daily growth rates after a one-year farming period (T7 = 353 days) for the different study groups reared in mid-water rope nurseries in a lagoon (LN) and reef (RN) habitat.

| Group | Nursery Habitat | Depth (m) | No. of Fragments | Stocking Period (days at T7) | Survival (% at T7) | Disease (% at T7) | Predation (mean% ± SE) | Fauna (mean% ± SE) | EV Size Augmentation (cm³ at T7 ± SE) | Daily Growth Rate (at T7 ± SE) |
|---|---|---|---|---|---|---|---|---|---|---|
| Acr_LN_5m | Lagoon | 5 | 48 | 353 | 81.25[*] | 17.95 | 0 | 22.62 ± 5.40 | 254.50 ± 35,23 | 0.02 ± .002[**] |
| Acr_RN_5m | Reef | 5 | 48 | 353 | 95.83[*] | 0 | 0.62 ± 0.67 | 3.87 ± 2.36 | 353.84 ± 38.23 | 0.04 ± .006[**] |
| Acr_total | all | 5 | 96 | 353 | 88.54 | 8.24 | 0.32 ± 0.35 | 13.24 ± 3.61 | 308.90 ± 26.72 | 0.03 ± .003 |

Significance levels: *** <0.001, ** <0.01 and *<0.05

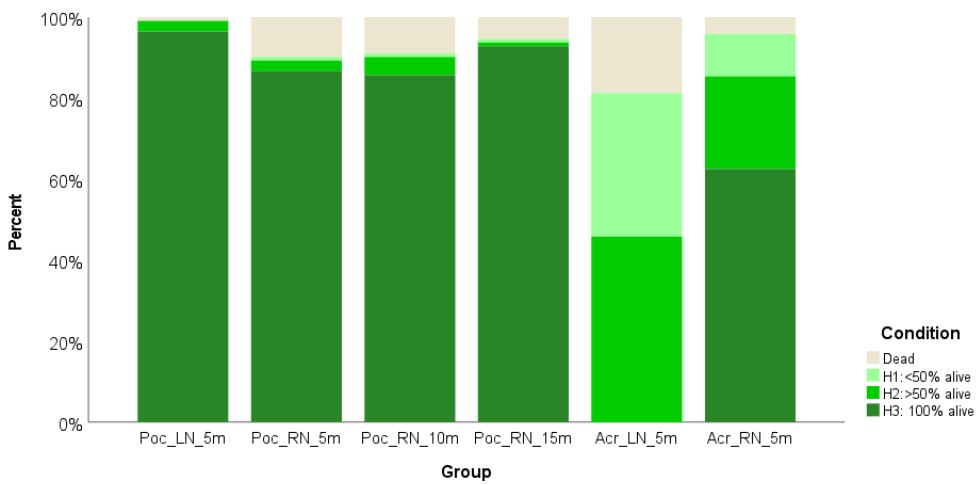

**Figure 3** **Condition of coral fragments after one year.** The figure shows four groups of *Pocillopora verrucosa* (Poc) and two groups of *Acropora* spp. (Acr) growing in a lagoon (LN) and a reef (RN) mid-water rope nursery at different depths for one year (T7).

## Growth

Fragment size was calculated as ecological volume (EV), which increased significantly for all groups during the one-year survey period (Fig. 4). The largest EV size increase (2195%) was observed in *P. verrucosa* fragments in the LN, which grew significantly from $41 \pm 2$ cm$^3$ to $905 \pm 31$ cm$^3$ in 371 days ($Z\,(N = 111) = -9.15$, $p < 0.001$; Fig. 4A). This was closely followed by fragments growing also at 5 m on the RN, which increased by 1957% (from $40 \pm 3$ cm$^3$ to $780 \pm 31$ cm$^3$; $Z\,(N = 101) = -8.72$, $p < 0.001$). The RN fragment size augmentation (Table 1) decreased with depth. At 10 m, the depth fragment increase was 1364% ($43 \pm 3$ cm$^3$ to $580 \pm 22$ cm$^3$; $Z\,(N = 102) = -8.77$, $p < 0.001$) while at 15 m the EV increase was reduced to a 1127% increase ($38 \pm 4$ cm to $390 \pm 40$ cm$^3$; $Z\,(N = 106) = -8.94$, $p < 0.001$).

Therefore, daily growth rates for *P. verrucosa* varied significantly between fragment groups ($F(3, 416) = 36.284$, $p < 0.001$). Post hoc testing revealed that there was no significant difference in daily growth rates between the lagoon ($M = 0.08 \pm .005$) and the reef ($M = 0.07 \pm .004$) at 5 m ($p = 0.848$). However, on the RN daily growth rates (see Table 1) varied significantly between the three rearing depths, with shallower depths showing faster growth rate ($p \leq 0.001$).

EV also increased for both *Acropora* spp. groups (Fig. 4B) during the one-year (353 days) farming period, in the LN by 738% (from $40 \pm 5$ cm$^3$ to $295 \pm 38$ cm$^3$; $Z\,(N = 38) = -5.37$, $p < 0.001$) and in the RN by 1098% (from $36 \pm 4$ cm$^3$ to $390 \pm 40$ cm$^3$; $Z\,(N = 46) = -5.91$, $p < 0.001$). Size augmentation and daily growth rates (Table 2) varied significantly between the LN and the RN at 5 m ($Z\,(N = 84) = 579$, $p = 0.008$), with fragments growing much faster on the reef.

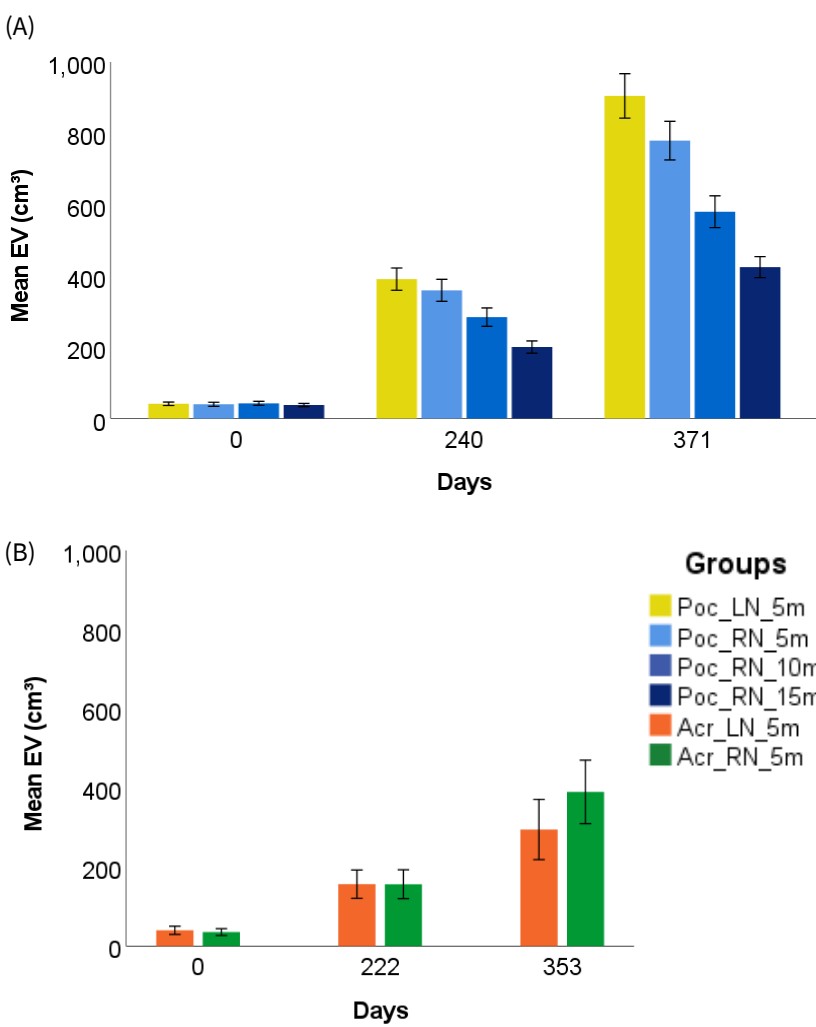

**Figure 4** **Coral ecological volume increase over one year.** The graphs show mean ecological volume (EV) at three different times (T0, T3, T7) during the one-year study period in a reef (RN) and a lagoon (LN) mid-water rope nursery for (A) *P. verrucosa* (Poc) fragments at different depths and (B) *Acropora* spp. (Acr) fragments at 5 m depth. Error bars: +/- 2 SE.

### Initial size

Average initial size at stocking for all *P. verrucosa* fragments was $5.22 \pm 1.1$ cm in diameter (h), ranged from 2.7 to 10.0 cm. A significant negative correlation between initial size EV and subsequent growth rate was found, with smaller fragments showing a faster growth rate ($r(418) = -0.56; p < 0.001$). This pattern was even more evident when analyzing treatment groups separately to account for the effect of depth (LN_5m: $r(109) = -0.65$; RN_5m: $r(99) = -0.65$; RN_10m: $r(100) = -0.63$; RN_15m: $r(104) = -0.68$; all $p < 0.001$; see Fig. 5).

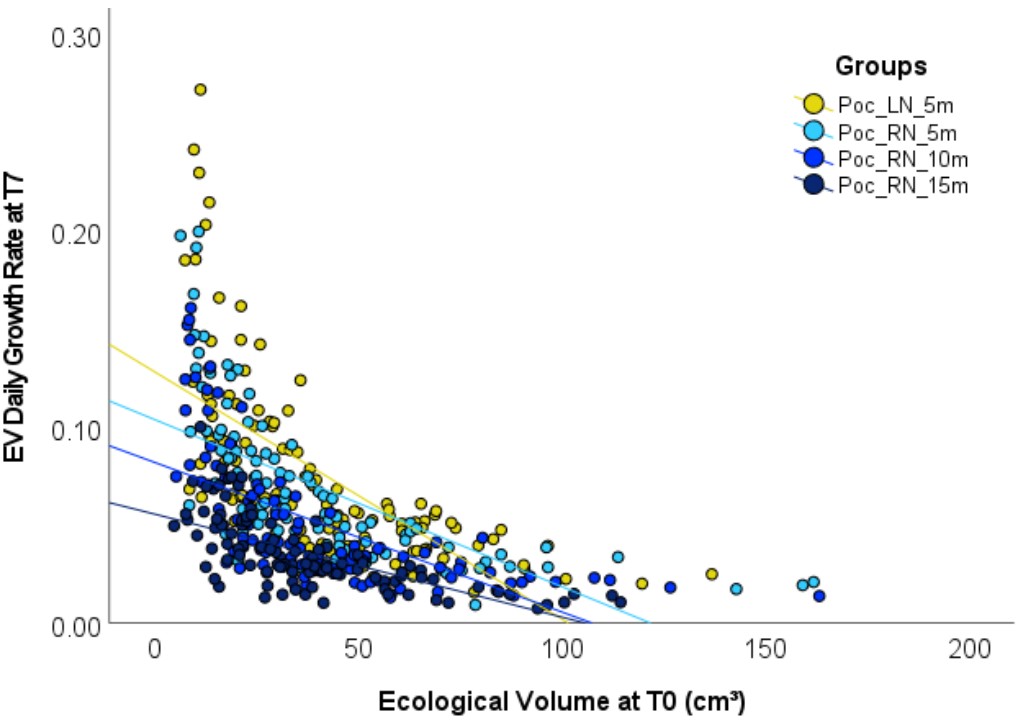

**Figure 5** **Correlation between *Pocillopora verrucosa* fragment stocking size and growth.** The scatterplot shows a significant negative correlation between fragment ecological volume (EV) at T0 and the EV daily growth rate at T7 ($r(418) = -0.56$; $p < 0.001$). A linear regression line was fitted for each group (LN_5m: $R^2_{Linear} = 0.42$; RN_5m: $R^2_{Linear} = 0.42$; RN_10m: $R^2_{Linear} = 0.40$; RN_15m: $R^2_{Linear} = 0.46$).

## Donor farming habitat

To investigate possible impacts of different donor farming habitats on fragments' growth rates in the two nurseries, the observed effect of initial size had to be controlled for first. Therefore, fragments from two reef farmed donor colonies with the two smallest mean stocking sizes as well as fragments from two lagoon farmed donor colonies with the largest stocking means were removed from the analysis. The remaining 141 fragments from 'reef farmed donors' and 156 fragments from 'lagoon farmed donors' were non-significantly different in stocking size at T0 ($Z$ ($N = 320$) $= 12419.5$, $p = 0.646$).

Growth rate comparison for these fragments at T7 revealed that *P. verrucosa* fragments that derived from reef donor colonies ($M_{Reef} = 0.0597 \pm .003$) grew significantly faster than fragments from lagoon farmed donor colonies ($M_{Lag} = 0.0478 \pm .003$) ($Z$ ($N = 297$) $= 8322$, $p < 0.001$). This was also the case when comparing daily growth rates for each study group separately (Fig. 6). In all but the RN_5m group fragments of reef farmed donors grew significantly faster than fragments that derived from lagoon donors, including the lagoon group (Poc_LN_5m), where fragments originating from reef farmed donors grew faster in the new habitat than fragments derived from lagoon farmed donor colonies ($Z$ ($N = 79$) $= 577$, $p = 0.047$).

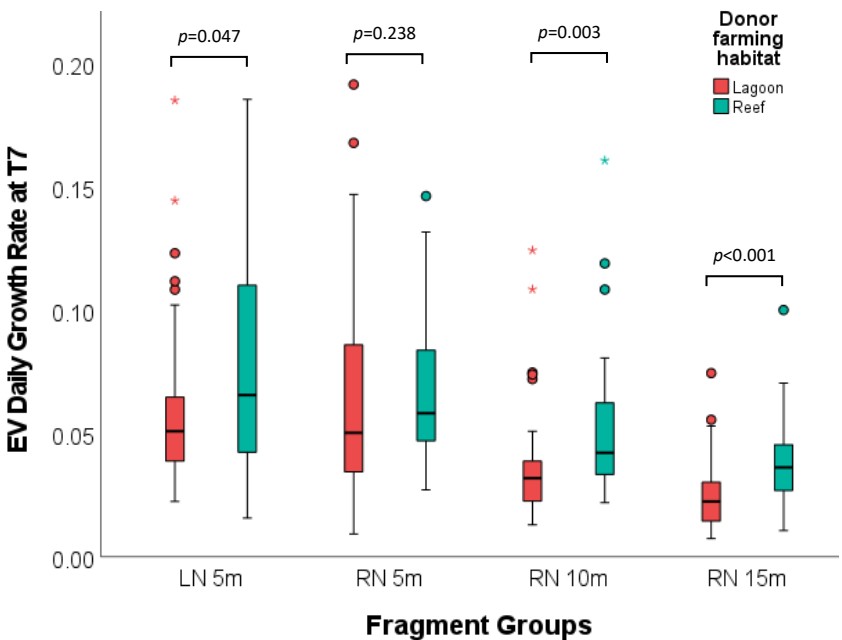

**Figure 6** Growth rate comparison of *Pocillopora verrucosa* fragments from different donor farming habitats for each study group. The boxplots show the comparison fragments within each study group (LN 5, RN5, RN10 and RN15), originating from different donors that were previously grown in either the 'lagoon' or the 'reef' farming habitat. Fragments derived from reef nursery reared donor colonies grew significantly faster in the lagoon nursery and at 10 and 15 m depth in the reef nursery.

## DISCUSSION

This study conducted a direct comparison and comprehensive assessment of mid-water rope nursery performance in a lagoon and a reef habitat in the Maldives for the first time. Our evaluations are based on fragment survival and growth as well as the occurrence of predation, disease and mutualistic fauna.

In both coral farming habitats, fragment survival was very high (81–99%) throughout the one-year study period. Similar survival rates have been reported, for example, from the Caribbean (85–96% for *Acropora cervicornis* after 12 months in in-situ nurseries; *Schopmeyer et al., 2017*) or the Philippines (96.4 ± 2.2% for *Pocillopora damicornis* after 10 months in a rope nursery; *Levy et al., 2010*). High fragment survival is critical for the success of the labor-intensive rearing phase of the coral gardening approach, so sufficient healthy colonies are available for the subsequent transplantation phase (*Edwards & Gomez, 2007*; *Frias-Torres, Montoya-Maya & Shah, 2018*). However, direct comparison revealed that *Pocillopora* fragments' survival was significantly higher in the LN, while *Acropora* fragments survived better in the RN. Closer inspection revealed that fragment survival and condition of both genera were affected differently by the spread of disease, which appeared to be coral genus and habitat specific, as only *Pocillopora* was affected on the reef while only *Acropora* was affected in the lagoon. For *Acropora* fragments, the negative effect of disease was also clearly noticeable when comparing growth, which was twice as

fast in the disease-free RN stock. These findings highlight the need to investigate coral diseases in coral restoration further, in particular possible transmission routes, time and density dependences in nurseries and mitigation measures. In this context, water quality and human induced pollution, in particular when operating in a resort setting, also require further attention. Disease outbreaks can significantly impact coral farming success and there is an additional danger of introducing disease to transplantation sites (*Moriarty et al., 2020*).

Coral predation is another factor that can hinder coral restoration success (*Miller et al., 2014*; *Koval et al., 2020*). Our study confirms that mid-water rope nurseries are very effective in keeping corals safe from known Maldivian corallivores such as the snail *Drupella* sp. or the starfish *Culcita* sp., which are regularly encountered in both habitats (*Montalbetti et al., 2019*; *Saponari et al., 2021*). All recorded predation incidents were from fish and hence they were more commonly observed on the reef, as one would expect. Nevertheless, predation scars were small and healed between survey intervals, therefore not directly impacting fragment condition. It should further be tested, if predatory fish occurrence could be reduced on the reef by placing the nursery structure further away from the safety of the reef slope (here only 5–10 m between structure and reef), if seafloor topography allows it.

We also investigated the occurrence of mutualistic fauna in the nurseries, in particular guard crabs, which can have positive impacts on coral health (*Glynn, 1987*; *Stewart et al., 2006*). The many benefits of hosting mutualistic fauna such as damselfish, decapods and hydrozoans have been widely studied, showing that it can reduce corallivory, sedimentation, predation, disease and even coral bleaching (*McKeon & Moore, 2014*; *Montano et al., 2017*; *Chase et al., 2018*; *Chase et al., 2020*). In line with these findings, our results suggest a positive correlation between guard crab presence and fragment survival. *Trapezia* sp. was first recorded in the coral stock after 8 months (T3), when fragments had reached a suitable size and branch complexity to host guard crabs. The percentage of fauna hosting corals was significantly higher in the lagoon, for both *Acropora* and *Pocillopora* fragments. There could be two, not mutually exclusive explanations for this observation. First, *Trapezia* sp. predators such as small reef inhabiting wrasses were never encountered during the surveys in the lagoon, while they have been regularly observed on the RN during maintenance work, which could indicate a higher predator abundance on the reef. In fact, increased predation pressure has previously been linked to reduced abundance of mutualistic decapods in *Pocillopora* colonies (*Stier & Leray, 2014*). Second, the LN hosted additional, older *Pocillopora* stock that was already populated by *Trapezia* crabs and hence population of the new fragments could have been facilitated. Movement of guard crabs between coral hosts to increase their reproductive success has been well documented (*Castro, 1978*) and deserves further attention. For instance, rearing fragments of mixed-age could be used to increase the abundance of mutualistic fauna and improve coral health in farming stocks.

Apart from coral survival, growth can be considered an important indicator of coral-farming success as it determines rearing time in the nurseries and therefore influences cost effectiveness and eventually restoration outcome (*Edwards, 2010*). Corals can reduce mortality risk by growing to a certain size (*Connell, 1973*; *Highsmith, 1982*), hence several

studies have looked at fragment size and depth as variables in coral nurseries (*Forsman, Rinkevich & Hunter, 2006*; *Soong & Chen, 2003*). Direct comparison between the LN and the RN showed that at shallow depth, *P. verrucosa* fragments grew at a similar rate indicating no apparent difference in farming environments. The insignificantly slower growth rate at 5 m in the RN is likely a result of the longer coral ropes that were pulled downwards (up to 7 m depth at the lowest point) as coral weight increased over time, even if this was counteracted with additional buoyancy devices. Although rearing depth had no effect on survival, *P. verrucosa* growth rates decreased by 27% from 5 to 10 and another 21% from 10 to 15 m on the reef as light levels decrease. Light availability is an important environmental parameter determining coral growth and typically reflected in the abundance of fast-growing corals in shallow depths (*Gladfelter, Monahan & Gladfelter, 1978*; *Grigg, 2006*) and the increased calcification rate in shallow waters (*Huston, 1985*), for which several mechanisms have been described (*Allemand et al., 2011*). The marked reduction in growth rate can be considered the main disadvantage over shallow farming locations such as lagoons. However, as it was the case in our study, the use of additional rearing levels at depth increased stocking capacity per nursery structure and could be an option to improve coral farming capacities and fragment output. Furthermore, the performance of outplanted colonies reared at different depths remains to be investigated.

To advance coral rearing success, fragment initial size should also be considered, although optimal size is likely species, method and location specific (*Edwards & Gomez, 2007*; *Edwards, 2010*). Our results for *P. verrucosa* in the Maldives indicate that smaller fragments grew significantly faster. We used an average stocking size of about five cm, with fragments ranging from 2.7 to 10 cm in maximum linear extension. In comparison, *P. damicornis* reared in rope nurseries in the Eastern Tropical Pacific exhibited a higher survival for fragments bigger than two cm but no significant difference in growth rate was found between size classes (*Ishida-Castañeda et al., 2020*).

Another interesting observation was that fragment genotypes deriving from 'reef-reared' donor colonies grew significantly faster in the RN as well as in the LN. One may expect that corals habituated to a particular environment may exhibit less stress after fragmentation if environmental conditions remain similar. Yet, we found that corals previously collected and farmed in the reef habitat generally outperformed fragments previously cultured in the lagoon, even after controlling for initial size. One noteworthy difference between donor colonies was initial rearing depth, which was generally deeper for 'reef-reared' donor colonies. *Pocillopora* is known to exhibit considerable environmental plasticity to adapt to variable conditions such as depth and water flow (*Soto et al., 2019*), but whether this could be a possible explanation for our observation and to what extent it is relevant to restoration practices remains to be further studied.

Finally, we observed some noteworthy points about nursery structure maintenance in our comparison of farming habitats. The removal of biofouling and sessile invertebrates typically constitutes a considerable workload and therefore cost factor in coral gardening (*Precht, 2006*).

Algae were observed growing over the nursery structures in both habitats, especially at shallow depths. In the RN, overgrowth decreased noticeably with coral growing depth,

likely as a result of reduced light, which reduces maintenance requirements. It has also been proposed that reef environments, home to a diverse community of herbivores and invertivores fish, can reduce nursery maintenance by providing a natural cleaning service and removing predators (*Gochfeld & Aeby, 1997*; *Frias-Torres et al., 2015*; *Frias-Torres & Van de Geer, 2015*). While this study did not intend to investigate the contribution of natural cleaning services, the five-month forced maintenance pause provided some useful insight. No significant damage or overgrowth of the fragments occurred in either habitat, except for some *Acropora* spp. fragments growing at 5 m on the RN, that were in part overgrown by brown algae.

It is also worth noting that the LN was placed further away from the island and daily resort activities, which impeded accessibility but did not prevent, for example, disease occurrence. In contrast, the RN was located along a popular diving and snorkeling route on the easily accessible house reef, therefore benefitting from increased public awareness and support for the project.

We limited our study to branching and fast growing *Acropora* and *Pocillopora* species, which are suitable and commonly used genera for this restoration method (*Levy et al., 2010*; *Mbije, Spanier & Rinkevich, 2010*). They are also promising candidates for restoring habitat complexity, considering that these key genera have been disproportionally affected by the previous mass-bleaching events (*Pisapia et al., 2017*). However, additional species should be included in the future to increase species diversity and therefore resilience of restoration sites.

Although our study site represents a typical resort island, situated in one of the most popular Maldivian atolls (*Statistical Yearbook of Maldives, 2020*), it should be considered that our findings are limited to a single location. Likewise, here we only assessed the first although important step of the coral gardening approach with research on the transplantation success of lagoon and reef reared corals to be conducted in the future. For instance, possible application advantages of reef rope nurseries for the transplantation phase could include more similar environmental conditions and shorter transportation to restoration sites.

Nevertheless, we hope to provide some new insight for restoration projects in the Maldives as such pilot studies are recommended to refine location and methods application (*Shaver et al., 2020*). In that way our study hopes to contribute by providing a sound assessment of mid-water rope nursery performance over a one-year study period in the Maldives and offers direct comparison of coral farming performance in a lagoon and reef habitat, which has not been conducted until now. As both nursery designs and habitats have been tested successfully, we suggest that Maldivian tourist resorts as well as local islands are suitable places for coral gardening projects, by the current standards of such endeavors and in a broader environmental management context (see *Hein et al., 2021*). Not only do they offer an opportunity to educate tourists and locals on the immediate threat this ecosystem is facing, they also offer a 'hands-on' approach in the face of seemingly overwhelming climate change threats. In parallel, such projects can help to draw attention to local disturbances, for example tourism overuse or pollution, which are more likely to get addressed in the context of a local awareness and restoration project.

## CONCLUSIONS

We conclude that reef and lagoon environments can provide suitable coral-farming habitats for mid-water rope nurseries in the Maldives, as our study demonstrated high survival and growth rates for *Pocillopora* and *Acropora* fragments over a one-year rearing period.

This provides a good starting point for the application of the coral gardening approach, although increased species diversity should be included as a restoration goal. We also found some habitat and genus-specific differences, that are worth considering in future restoration projects. In direct comparison, the robust *Pocillopora* fragments performed better in a lagoon habitat and were less impacted by disease, while *Acropora* rearing success was better in the reef habitat. Smaller initial size (<5 cm) at stocking increases growth rates for *Pocillopora* in both habitats, while increased rearing depth decreases fragment growth. We suggest that mutualistic fauna, here more abundant in the lagoon, could be increased by stocking fragments together with older colonies to facilitate transmission. Furthermore, apart from fish predation, our mid-water rope nurseries provided good protection from corallivory in the lagoon and reef habitat. How different farming habitats and rearing depths translate into outplanting success of coral gardening remains to be tested.

Finally, we consider reef mid-water rope nurseries a useful addition to the coral restoration tool kit in the Maldives, especially when lagoon farming habitats are not available, not easily accessible or conditions are unsuitable. Our streamlined rope nursery design withstood the high currents and fish abundance in the reef environment, while providing additional rearing space at depth. Therefore, we hope that this novel research provides some valuable insights for restoration practitioners and a step towards expanding restoration efforts in the Maldives.

## ACKNOWLEDGEMENTS

The authors would like to thank Planhotel Hospitality Group, IDive Maldives, Athuruga Island watersports team and the families Trotman and Steiner for their support.

### Funding
The authors received no funding for this work.

### Competing Interests
The authors declare there are no competing interests.

### Author Contributions
- Inga Dehnert conceived and designed the experiments, performed the experiments, analyzed the data, prepared figures and/or tables, authored or reviewed drafts of the paper, and approved the final draft.
- Luca Saponari performed the experiments, authored or reviewed drafts of the paper, and approved the final draft.

- Paolo Galli analyzed the data, authored or reviewed drafts of the paper, and approved the final draft.
- Simone Montano conceived and designed the experiments, prepared figures and/or tables, authored or reviewed drafts of the paper, and approved the final draft.

## Data Availability

The raw data is available in the Supplemental File.

## Supplemental Information

Supplemental information for this article can be found online at http://dx.doi.org/10.7717/peerj.12874#supplemental-information.

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
