# Peer review of "Comparing different farming habitats for mid-water rope nurseries to advance coral restoration efforts in the Maldives"

_PeerJ, doi:10.7717/peerj.12874_

## Round 0.1 · original submission · Major Revisions

I have heard back from two reviewers, both of whom had positive things to say about your work while providing constructive comments on how to improve the paper. I have read your paper and their comments, and find them fair and appropriate. I look forward to seeing your revised version.

Reviewer 1 ·

Basic reporting

This paper describes a straightforward study articulating performance of coral fragments of two taxa in coral nurseries in two habitats. The results show, somewhat expectantly, contrasting performance of the two taxa in the two habitats. The study also includes more extensive quantification of ecological interactions (predation, disease, algal overgrowth, guard crabs) than is common in nursery studies.

Experimental design

These results thus do indeed provide some helpful basis for planning additional coral gardening activities in the region. However, I would say that the appropriate guidance derived from this study is greatly limited by the lack of any information on the post-outplant performance of these gardened fragments. In the end, the survival and growth of the frags on the actual reef is what we care about, and other studies have shown that this may or may not be closely predicted by the frags’ performance in the nursery setting. The authors acknowledge this point very briefly in the discussion, but the usefulness of the current results is greatly limited by its lack.

Validity of the findings

No comments

Additional comments

Two other points I would suggest substantive revision; the first is semantic. Although the term ‘nursery’ refers to a wide range of settings related to infant or juvenile organisms, the term ‘nursing’, at least in US English, is used primarily to refer to breastfeeding of human infants. Therefore, I suggest to remove the term ‘nursing’, ‘nursing period’, ‘nursing environment’, ‘nursing performance’ etc. from the paper. (‘nursery’ can be replaced in most cases).
The second relates to a general expectation about corals and habitat performance. Existing best-practice for coral gardening suggests there is probably benefit in matching the nursery habitat to 1) the habitat from which corals are provenance and/or 2) the habitat that is intended to be restored. In fact, there is also benefit from sourcing restoration genotypes from across a diverse habitat range (to incorporate as much adaptive genetic variability as possible – see Baums et al. 2019 Considerations for maximizing the adaptive potential of restored coral populations in the western Atlantic. Ecol Appl. 29:e01978 https://doi.org/10.1002/eap.1978) and this provides an important framing for testing habitat-specific nursery performance as addressed in the current paper. However, the authors discuss and analyze the effects of ‘colony origin’ in the results, by which they mean the prior nursery culture location of the colonies. If there is a local habitat adaptation in these populations, the actual origin of the genotypes (i.e. what reef habitat were they originally collected from) may be important. Where were these genotypes originally from? And what effect might this actual (collection) origin have on differential performance between habitats.
Minor corrections:
Ln 31: the different descriptions of number of fragments seem inconsistent throughout. Here in the abstract, it says ‘eight fragment replicates from the same donor colony (N=20) in each group’. This sounds like 32 fragments were made from each donor colony, but then what does ‘N=20’ mean?

Ln 173: here in the methods it would be nice to give a bit more info on ‘predation’ and ‘associated organisms’. There is more info given in the discussion, but as a reader, I was curious what evidence and source of predation you were scoring (scars, bite marks, active feeding?)

Ln 221-222: Clarification: This sentence says that all but 3 of these fragments suffered more than 50% mortality – I think this is not correct(?). Also, delete redundant ‘in’

Ln 251: Temporarily should be Temporary

Ln 280-281: Mh appears for the first time. Should be defined in the Methods. Also, the term ‘longest linear extension’ is ambiguous. I think you mean simply ‘colony diameter or ‘colony dimension’ either of which would be less ambiguous terms. (Other studies use the term ‘total linear extension’ to indicate the total branch length in a branching colony, rather than the colony dimension)

Ln 336: I believe that Moriarty is missing from the Refs cited section.

Ln 340: Save should be safe

Ln 402: extend should be extent

Ln 418: the meaning of the phrase ‘was placed offsite resort activities’ is unclear. Does this just mean the placement of the nursery was further distant and not affected by resort activities? Should be re-phrased.

Discussion/Conclusion sections: generally, could be tightened/shortened. These sections completely recapitulate all the results and are at times repetitive.

Reviewer 2 ·

Basic reporting

In general, the study demonstrate the application of mid-water rope coral nurseries in Maldives and found both habitats (lagoon nursery and reef nursery) are suitable for coral restoration using the proposed method. The manuscript is easy to understand. However, there are several modification need to be done before accepting the manuscript as indicated in attached notes.

Experimental design

Author provide a well defined research and within the scope of the journal. Require minor modification by adding some details related to the location of the study, sample collection method, etc. See details in attached document.

Validity of the findings

No major comment. See details in attached document.

Additional comments

Even though the mid-water rope coral nurseries have been applied in other places, this study conducted a preliminary test on different reef habitat to seek efficacy of the method for coral restoration. However, several details need to be added (as indicated in attached document) to improve the quality of the manuscript. The study is suggested for publication due to the important of coral restoration in tackling the coral reefs degradation.

Annotated reviews are not available for download in order to protect the identity of reviewers who chose to remain anonymous.

---

## Round 0.2 · Minor Revisions

Your paper is almost ready to be accepted, but as one reviewer has noted, the colony origin section remains a bit hard to follow. Please check these last comments, and I look forward to seeing a revised version soon.

Reviewer 1 ·

Basic reporting

This revision is indeed improved. Although the new paragraph (~Ln 159-168) adds information, it is still difficult to understand the history of the colonies used in the experiment and the potential effect of the natal habitat. As I understand it, Pocillopora colonies were collected in 2018 from two separate sites. One (off-island) collection site populated the LN and one the RN (what kind of ‘artificial structure’ were they collected from?). Then, two years later, these colonies were reciprocally mixed between the two nursery habitats such that there were equal colonies from both ‘collection habitats’ in each ‘nursery habitat’ treatment in the study reported here. Thus, the conclusion that the RN ‘nursery habitat’ treatment had higher growth rates (clearly an interesting result) includes genotypes that were originally collected from two separate (natal) habitats that were subsequently cultured for two years in different habitats. Is this correct? Was there any difference in the performance of the genotypes from the two different (natal) habitat origins? Perhaps a supplemental figure depicting the steps in the history of the experimental colonies would be helpful?
The language is also still cumbersome in places. Example (Ln 430-432) “Yet, we found that reef farmed donor fragments outperformed fragments from lagoon reared donor corals in growth rate in nearly all cases, even though stocking size was controlled for.” I believe that ‘reef-farmed’ and ‘lagoon-reared’ should be hyphenated in this construction, but it would perhaps be much simpler to say ‘Fragments previously cultured in the RN outperformed those previously reared in the LN even after controlling for initial fragment size.’ Also, I think that ‘initial size’ is a better term than ‘stocking size’ . . .
Example (Ln 441) “ . . Structure over-growing algae were found in both habitats . . “ would be clearer to say ‘Algae were observed fouling (or growing over) the nursery structures in both habitats’ Example (Ln 504) “the newly adapted, streamlined design of the reef rope nursery passed its test in a current and fish rich environment” Clearer to say: “Our streamlined rope nursery design withstood the high currents and fish abundance in the outer reef environment . . .”

Ln 496: This is where better acknowledgement of the consideration of the destination (outplanting) habitat is needed. It is premature to recommend prioritizing shallower depths for nursery culture unless and until performance in appropriate outplanting destinations is demonstrated. If shallow outplanting sites are intended, this is probably true, but may not be true if deeper outplanting is intended.
Moriarty et al. is still obscured in the refs section
Fig 1 A seems to have 3 separate scales (though only one is given) and it should be articulated what is represented in depiction of the atoll (I assume gray is land and white is ocean? What is the black region?)
Fig 6: Easier to interpret by placing p-value on the figure over each pair of bars (not in the legend)

Experimental design

As above, still challenging to understand the origin of the colonies involved and potential influence of genotype origin

Validity of the findings

Fine

Reviewer 2 ·

Basic reporting

The author follow all the comments suggested by the reviewers by providing a clear and meaningful explanation on the comments stated previously. Language used is clear and much easier to understand by reader. The authors also add details on literate review regarding the coal reefs status in Maldives and several successful coral restoration program to support certain statement. Figures were improved by adding photos which will help readers to understand more on the situation of the study. The authors also issued on positive and negative implications of coral farming environments and their specific challenges, which is important consideration for future study.

Experimental design

The study following the aims and scope of the journal which is appropriate in the Biological Sciences and Environmental Sciences section. The research question is well defined, relevant and meaningful for coral restoration program, particularly in Maldivian coral reefs as the research related to coral restoration is still scarce in the region. Description in the method was improved and provide sufficient information on the study area and coral restoration technique applied in the study.

Validity of the findings

The mid-water coral nursery have been applied as part of the reef management strategy to overcome the coral degradation. Even though the research on the coral restoration using the mid-water rope have been well described in other studies, the present research will add value to the literature and helps the reef managers to understand and plan the coral restoration strategy for conservation purposes. The findings presented in the manuscript are statistically tested and well explained. Conclusion is well described with some recommendation for future studies.

Additional comments

Overall, the authors followed all the recommendation and suggestion by the reviewers which improved the quality of the manuscript in line with PEERJ standard. As a second reviewer, I'm satisfied with all the changes made in the manuscript and the authors answered all the comments appropriately.

---

## Round 0.3 · accepted · Accept

Thank you for your revision; I am pleased to accept this work and look forward to seeing the published paper! Congratulations!